# Blood Transfusion in Equids—A Practical Approach and Review

**DOI:** 10.3390/ani12172162

**Published:** 2022-08-23

**Authors:** Camilla A. Jamieson, Sarah L. Baillie, Jessica P. Johnson

**Affiliations:** Equine Veterinary Medical Center, Member of Qatar Foundation, Education City, Doha P.O. Box 6788, Qatar

**Keywords:** horse, emergency medicine, transfusion, hematology, critical care

## Abstract

**Simple Summary:**

Transfusion medicine is an accessible, technically simple, and often lifesaving tool that can be used in both field and hospital settings, in cases of significant bleeding or anemia. A thorough understanding of the indications, methodology and complications of blood transfusion allows the practitioner to identify cases where administration of whole blood is necessary, and how to safely perform the transfusion. This review collects the current literature surrounding blood transfusion into one readily accessible document to allow clinicians a comprehensive understanding of all aspects of equine blood transfusion, while serving as a reference for performing these procedures.

**Abstract:**

Transfusion medicine is a crucial part of equine intensive and critical care. Blood transfusions can save lives in both acute and chronic cases of anemia, hemorrhage, and hemolysis. It is vital to have a comprehensive theoretical and practical understanding of the techniques, implications, risks, and complications. This review covers the physiology and pathophysiology of conditions requiring transfusion, as well as step by step guidance for practitioners of all experience levels. This review is designed to serve as a practical reference for those who are treating horses in either the field or hospital setting. It aims to provide both theoretical background and easy to locate formulae with guidance that is easy to refer to in a critical situation. When risks and benefits are well understood, these techniques can be confidently employed in critical situations to improve outcomes and save lives.

## 1. Introduction

Blood transfusion is a lifesaving procedure in critical care settings across all species and is often required in equine intensive care and emergency situations. There are numerous reasons why horses may require a transfusion, and when administered thoughtfully, at the appropriate time and in the correct way, blood or blood component transfusion can be one of the most readily accessible lifesaving procedures that a veterinarian can perform. Despite this, there are certain risks and safety measures that the clinician must be aware of and understand how to mitigate, before embarking on blood transfusions, to optimize the outcome for the patient while reducing the incidence of unpleasant to fatal adverse reactions. 

Circulating blood is made up of erythrocytes suspended in a milieu of extracellular water, electrolytes and biologically active proteins [1]. These components each play a key role in tissue oxygen delivery, coagulation, all aspects of immune functions, maintenance of oncotic pressure and osmolarity, and cellular function and signaling [2,3,4]. Due to the complex nature of the blood above and beyond simply delivering oxygen to tissues, the implications for blood transfusion are numerous and the nature of adverse events associated with transfusion can also vary. The equine spleen has a species-specific ability to store a vast reserve of erythrocytes [5]. Horses respond to increased oxygen demand by sympathetic vasomotor contraction of the spleen to mobilize sequestered erythrocytes into circulation, in response to both acute and chronic demand. In the most extreme circumstances, an adult horse can double their PCV with splenic contraction [5,6,7], and this must be taken into account when evaluating PCV, particularly in the face of acute blood volume changes. 

## 2. Disorders Requiring Transfusion 

### 2.1. Whole Blood Loss

Pathological process resulting in a rapid drop in packed cell volume (PCV), whole blood loss with rapid circulating volume depletion even before a drop in PCV is noted, or slow decline in PCV over time to below a critical threshold, often result in the need for blood transfusions to replenish lost erythrocytes and restore the oxygen-carrying capacity of the blood [8]. Horses require whole blood or packed red cell transfusion when the erythrocyte count, and, therefore, the blood oxygen delivery capacity, drops below the threshold at which increased tissue oxygen extraction can compensate for decreased oxygen delivery [9]. Whole blood loss can rapidly result in the need for transfusion, to replace circulating volume and maintain tissue perfusion.

#### 2.1.1. Traumatic Blood Loss

Traumatic external whole blood loss anemia may be the easiest to quantify, as the clinician may be able to visually estimate the volume of blood lost and estimate the volume depleted. The time of onset and origin of blood loss may be apparent, helping the clinician to accurately estimate lost volume, and hopefully, ongoing losses can be curtailed. In human critical care, hemorrhage is graded I-IV, based on percentage blood loss with treatment criteria and evidence for prognostication associated with each grading [10]. Hemorrhage resulting in loss of a significant percentage of circulating volume is relatively straightforward to understand, as the same proportions of all blood components are lost at the same rate [8]. A cause of significant, often fatal hemorrhage in equids specifically, is carotid arterial hemorrhage secondary to mycotic infection of the guttural pouch [11]. Whole blood loss can occur into body cavities as well as externally, with the abdomen being the most common site, followed by into or around the reproductive tract and/or into the thorax [12,13,14]. Broad ligament hemorrhage can be a life-threatening occurrence in periparturient mares requiring emergency treatment, with lesions occurring most commonly in the proximal uterine artery [15,16]. Certain surgeries in vascular regions are associated with acute hemorrhage, specifically surgery of the sinuses, ethmoid turbinates and vascular structures [17,18], which are discussed in greater detail in the surgical hemorrhage section. Whole blood loss can also be due to hemostatic disorders in the face of usually insignificant trauma [3,19,20].

#### 2.1.2. Surgical Blood Loss

Certain surgeries are associated with an increased risk of both intra- and postoperative hemorrhage. As mentioned above, these include surgeries involving the head and paranasal sinuses, reproductive tract and spleen, as well as the foot, and various foreign-body and tumor removal surgeries [21,22,23,24,25,26]. Intraoperative hemorrhage is deemed to be a surgical complication if it is unexpected and/or severe enough to warrant blood transfusion [27]. It can hamper the surgical field of view and result in a number of sequelae, such as anemia, postoperative seroma formation as the free blood within the surgical site coagulates and the clot contracts leading to a seroma, and surgical site infection. Even mortality induced by hemorrhagic shock has been reported [22,27]. Careful preoperative screening prior to elective procedures, as well as proper surgical planning, can help to reduce the risk of complications associated with intra- and postoperative blood loss [22,27]. A thorough history-taking may highlight pertinent features such as recent use of non-steroidal anti inflammatory drugs (NSAIDs) or other drugs known to impair coagulation [27]. Physical examination may highlight potential issues such as pallor or petechiation, or other findings, which could prompt clinical pathologic assessment. Careful attention should be paid not only to hematologic values, but also to liver enzymes [27]. If pre-anesthetic screening indicates, further coagulation testing should be performed. In elective cases, the option exists to delay the surgery to facilitate investigation and treatment, or at least, to allow for sufficient preparation. 

Where a patient is identified as being at increased risk of intraoperative hemorrhage, attention should be paid to positioning. For example, where paranasal sinus surgeries are performed under general anesthesia, the horse can be positioned in reverse Trendelenberg (elevating the region being operated on relative to the heart, to reduce perfusion and, therefore, reduce hemorrhage) to reduce blood loss [27]. Depending on the surgical site, tourniquets can be applied if blood loss from the distal limb is anticipated [24]. Good anatomical knowledge of the intended surgical site is essential to identify and isolate major blood vessels in order to avoid iatrogenic damage to such vital structures which could result in hemorrhage, as well as employing best surgical practice in line with Halsted’s principles, which are the seven pillars of excellent surgical technique, including gentle tissue handling, meticulous hemostasis, obliteration of dead space, asepsis, preservation of blood supply, gentle tissue handling and accurate apposition of tissues [28]. Surgical preparedness is essential for responding to, and minimizing, hemorrhage where it occurs [28]. Where intraoperative hemorrhage occurs, efforts should be made to record the volume of blood lost, to monitor on-going losses and facilitate later transfusion calculations if necessary. Where hemorrhage occurs, surgical hemostasis techniques can include mechanical hemostasis, such as the application of pressure and use of ligatures and vascular staples [28]. Thermal hemostasis using electrosurgery, and chemical hemostasis, such as use of topical epinephrine to achieve vasoconstriction, are commonly used. Various sponge materials are available (e.g., gelatin foams, celluloses and collagen) which can be applied topically to wounds to provide a scaffold for clot formation [28]. In cases of bleeding from bone, topical bone wax can also be applied [28]. Where surgical hemostasis techniques fail, medical options are the next logical step. Antifibrinolytic lysine analogues (e.g., aminocaproic acid and tranexamic acid) act by inhibition of clot breakdown, and can contribute to achieving hemostasis in emergency situations [28]. The decision for blood transfusion should not necessarily be left to the final option. Several factors should be considered before deciding to perform a blood transfusion, as outlined below. 

#### 2.1.3. Hemostatic Disorders

Abnormalities in any aspect of the coagulation cascade can lead to excessive bleeding in response to usually unremarkable levels of trauma [29,30]. Disorders of hemostasis can be genetic or acquired. Genetic/heritable coagulopathies include Won Willebrand’s disease, Hemophilia A, inherited Prekallikrein deficiency, and Glanzmann’s and atypical Thrombasthenia [29]. These conditions are characterized by a heritable lack of production of specific proteins in the coagulation cascade. Acquired coagulopathies have a broader range of inciting causes. The two most common are disseminated intravascular coagulation (DIC) secondary to endotoxemia, and purpura hemorrhagica (PH). PH is a severe complication of *Streptococcus equi* var *equi* infection, vaccination and rarely *Corynebacterium pseudotuberculosis* infection [31]. In all these cases, the horse’s ability to clot adequately is decreased, which may result in excessive and hard to control hemorrhage.

### 2.2. Anemia 

Anemia is defined as a decrease in erythrocyte concentration in the circulating blood. Anemia can be acute or chronic, can have a varying degree of clinical significance, and has an extensive list of differentials. Anemia can be initially divided into two categories, decreased red blood cell production, and increased red cell loss [19]. Post hemorrhage anemia occurs in the 3–28 days following blood loss, when the plasma volume has returned to normal, but the erythrocyte concentration has not yet rebounded [10].

Anemia attributable to decreased erythrocyte production is either primary, due to immune-mediated or neoplastic destruction of the bone marrow, or secondary, due to a systemic, extra-hematopoietic-system insult. Primary bone marrow destruction may be transient or permanent, due to immune-mediated or neoplastic conditions. Secondary bone marrow suppression is due to chronic disease, absolute iron deficiency, viral infection, plant or medication toxicosis, or heavy metal toxicity [3,19,20]. 

Anemia caused by increased destruction of erythrocytes is classified as either intravascular or extravascular. Intravascular hemolysis occurs when erythrocytes are lysed and destroyed within the vascular space. Extravascular hemolysis leads to anemia when an excessive number of erythrocytes are removed from circulation, by macrophage populations in the spleen, liver and bone marrow. Extravascular hemolytic anemia uses physiological erythrophagocytic mechanisms, with excessive efficiency usually due to changes in the structure, antigenic expression or pliability of the erythrocyte membranes, inducing increased elimination from circulation and breakdown [32]. 

The hallmark of intravascular hemolysis is the lysis of erythrocytes within the circulating space, causing discolored serum, hemoglobinemia hemoglobinuria [33]. Mechanisms inducing intravascular hemolysis are varied but include toxin ingestion, envenomation particularly by rattle snake, metabolic disturbances, intracellular parasitemia, and iatrogenic administration of fluids of improper tonicity [34,35,36,37], but causes can be varied and individual.

#### Anemia of Chronic Disease

Anemia of chronic disease is one of the most common causes of anemia in horses, with a complex pathophysiology. In horses, as in people, chronic inflammation leads to decreased naturally occurring erythropoietin (EPO) production and simultaneously attenuated bone marrow response to EPO, while chronic inflammatory conditions increase erythrocyte breakdown due to increased hemophagocytosis by macrophages. This leads to sequestration of iron within the phagocytic cell populations and depletion from the erythrocyte precursor cell lineages, while inflammation increases hepcidin concentrations, further sequestering biologically-available iron stores [38,39]. The use of recombinant human erythropoietin (rhEPO) is a controversial topic in equine medicine, as it has been used in some cases in an attempt to increase red cell volume, and, therefore, oxygen carrying capacity in performance horses [40]. It is, of course, banned in that capacity for competition horses, and has been associated with fatalities, erythroid hypoplasia, and anemia, so it is not advised for use in equines [41]. rhEPO is now frequently screened for in anti-doping labs, using specific antigens against the recombinant human proteins in equine plasma [42] and is not recommended for use in clinical cases of anemia, instead, addressing the underlying issues is recommended, and allowing the horse to recover PCV naturally.

### 2.3. Other Disorders Requiring Transfusion

There is a subset of cases where whole blood transfusion may be beneficial, even when the PCV is normal. If blood components are not available to the practitioner, administration of a whole blood transfusion may provide lifesaving clotting factors [30], albumin, and plasma proteins to restore oncotic pressure to the horse. Additionally, in rare circumstances such as carbon monoxide poisoning, transfusion may be required to provide erythrocytes with oxygen carrying capacity even when the absolute erythrocyte count is normal [43]. 

## 3. Indications for Transfusion

The need for blood transfusion is based on either the horse exhibiting clinical signs of hypoxemia, hypovolemia and decreased tissue oxygen delivery, or estimated loss of more than 20% of the individual’s circulating blood volume [44]. There is no one single criteria that a clinician can use to determine the need for blood transfusion, rather the clinician must consider the constellation of clinical and clinicopathological data presented to them, in combination with historical information and experience, to determine if and when a blood transfusion is indicated. The information presented here may help the clinician to develop a decision-making rubric to apply in each case where they are considering administering a blood transfusion. The more rapidly blood is lost, the more likely the animal is to require transfusion, as the end organ tissues have less time to adapt their oxygen demands and extraction efficiency, and acute whole blood hemorrhage impairs both the oxygen carrying capacity of the blood as well as the circulating volume and, therefore, perfusion pressures. The need for transfusion can be quantified by calculating the oxygen extraction ratio (O_2_ER = VO_2_/DO_2_) where VO_2_ is the venous partial pressure of oxygen and DO_2_ is the arterial or direct partial pressure of oxygen. This ratio is a formula for calculating the percentage of the oxygen that is delivered to the target tissue, that is taken out of the blood. In normal mammals, the oxygen extraction is around 30% and the additional 70% of oxygen is recirculated. As the oxygen carrying capacity of the blood is lost (among other reasons such as hypoxia) the oxygen extraction percentage will increase. This indicates that the tissues are removing more than the normal percentage of oxygen from the blood. As oxygen extraction approaches 60%, oxygen sparing mechanisms are activated in the body and shunting of blood to preserve essential organ function occurs [4,45,46]. Determining the need for acute blood administration in the face of hemorrhage or severe acute blood loss is often a more straightforward decision as the life-threatening nature of the decreased oxygen carrying capacity is readily apparent to the clinician. The indications for transfusion in cases of chronic anemia may be less readily apparent, and must focus on indicators of inadequate tissue oxygen delivery. 

Clinical signs indicating the need for transfusion include tachycardia, tachypnoea, decreased pulse quality, cool extremities, pale mucus membranes, mentation changes (anxiety, distress, or shock/depression), hyperlactatemia and decreased PCV with or without decreased total solids, as detailed in Table 1 [8]. Compulsive thirst is something often observed in the acutely hemorrhaging horse, as sudden drops in central nervous system perfusion lead to activation of the central thirst centers and rapid upregulation of the renin–angiotensin–aldosterone system [47,48]. Changes in PCV can lag behind the clinical requirement for blood in cases of acute blood loss, since whole blood loss depletes erythrocytes, serum and total solids (TS) in equal proportions, and the horse has a unique ability to increase PCV via splenic contraction. In the case of acute severe blood loss, all components of the blood are lost in equal proportions. As the horse is bleeding, circulating volume drops, but proportions of blood components have not changed. As the horse stabilizes, either spontaneously by shifting intracellular fluid to the extracellular space, or with the administration of procoagulants and intravenous crystalloid fluids, the total protein and PCV will drop significantly. The equine ability to contract the spleen in response to epinephrine stimulation and hypoxia additionally confounds interpretation of PCVs in the face of acute trauma. Splenic contraction can increase the PCV by up to 30–50% in the acute phase and it takes 12–36 h for the PCV to return to baseline [6]. 

## 4. Practicalrfttttities of Transfusion

### 4.1. Donor Selection

Once the requirement for blood transfusion has been identified, the clinician must select an appropriate donor. A small number of institutions maintain herds of blood donor horses, in closed herds. These horses are blood-typed, routinely monitored for infectious diseases, and who are tolerant of the stresses associated with the blood donation procedure. In the absence of a herd of donors, a large (>350 kg), skeletally mature gelding in good health, with a good temperament, is ideal. Utmost effort should be made to acquire a minimum database of hematology, serum biochemistry, and confirm negative Equine Infectious Anemia status; this is the ideal, however in the most critical of cases, donor selection may be limited and the risks of transfusion are weighed against the urgency of the blood requirement. In only the most critical of situations should an unscreened donor be considered, and all samples should be collected and submitted to the lab so the clinician and owner can be aware of all risks assumed. If a gelding is not available, a mare or stallion who has no reproductive history can be used as a suitable alternative [49].

### 4.2. Blood Donor Disease Screening

Screening of blood donors in a commercial or university setting should be extensive and carried out at least annually, and every 6 months for diseases endemic to the region. Screening should include testing for Equine Infectious Anemia (EIA), the equine hepaciviruses, and Equine Viral Arteritis (EVA) [49]. In regions where Brucellosis Piroplasmosis, Glanders and Dourine are endemic, these should be included in screening protocols [50]. Additionally, it is pertinent to perform hematology, serum biochemistry, and inflammatory marker screenings at regular intervals. If the possible, blood typing the donor herd is an excellent way to help expedite the transfusion process when needed. When a pre-screened donor herd is not available, as is often the case, a donor who is up-to-date on vaccines, with a current negative EIA test, and who resides at the same property as the recipient, is recommended. The authors recommend using horses owned by the same person, if possible, to minimize liability in the event of transmission of any communicable disease. In specific cases such as Neonatal Isoerythrolysis, the clinician has two choices for selecting a donor, either the dam may be used, and her red cells washed (see Section 7), or a donor gelding with a compatible cross match can be employed.

### 4.3. Biological Products and Hepacivirues

Aside from iron overload hepatitis, there are a number of hepatotropic viruses that can be transmitted in equine blood products. These include Equine Hepacivirus and Equine Parvovirus, the cause of equine serum hepatitis. It has been shown that these viruses, particularly Equine Hepacivirus, can be transmitted through plasma, serum, tetanus antitoxin and other biologic origin equine products [51,52]. While viral screening is commonplace in commercially produced equine biological products, it is often not possible to wait for testing of donors before transfusions are administered. However, if blood is being stored for later use, or a donor herd is being established, potential donors should be screened for equine hepatitis viruses. Additionally, horses with liver disease occurring in the months following administration of biological products should be screened for the new hepatic viruses.

### 4.4. Cross Matching and Blood Typing

#### 4.4.1. Blood Typing

There are seven blood group systems in horses, A, C, D, K, P, Q and U. Within these blood groups there are several specific typing antigens, or ‘factors’ expressed on erythrocytes, which are denoted with lowercase letters. The specific blood type of an individual is denoted with an uppercase letter and a lowercase letter, specifying the blood group and type of the animal [53,54,55]. These seven systems with multiple sub-allelic variations permute to give over 400,000 possible individual blood types for horses. For a donor and recipient to be truly compatible the system and antigen must be the same, but there is a spectrum of reactivity within the blood groups. A number of commercial labs around the world offer blood typing services, which is an excellent option for those wishing to establish a donor herd. There is one available stall side test kit, the Alvedia QuickTest Equine blood typing kit, to determine the presence or absence of Ca blood group antigens, which are considered the most allogenically reactive [56].

#### 4.4.2. Cross Matching

Cross matching is performed to determine immunological reactivity between two individual samples from two different horses. Horses can have different blood types but be cross-match compatible. Cross matching has historically required laboratory reagents that are not readily available to the practitioner, however there are stall-side agglutination tests which give the clinician insight into reactivity of donor and recipient blood fractions, and there is now a stall-side cross match kit on the equine market, the Alvedia GelTest Equine Cross Match, which is validated with blood type dependent agreement with laboratory references [57].

The major cross match identifies compatibility between the recipient serum and the donor erythrocytes, and gives the clinician information about both chances of transfusion reaction, and survival time of the erythrocytes administered [56]. The minor cross match assesses immunologic reaction between donor serum and recipient red blood cells. In the absence of a lab or stall-side cross match, the clinician can perform a slide agglutination test (SAT). While the SAT has been superseded by significantly more accurate testing such as lab or stall-side cross matching, it can give the field clinician some confidence that donor and recipient blood is compatible [58], particularly in the face of multiple transfusions. This can be performed by placing one drop of donor serum, one drop of recipient red cells, and one drop of normal saline on a glass slide, gently mixing the three components and assessing for agglutination, see Figure 1.

### 4.5. Other Tests

Cross matching and blood typing are the most accurate laboratory methods to assess blood product compatability. Other tests that are often discussed when transfusing horses are the Coombs Test and the jaundiced foal agglutination test (JFAT). The Coombs test is a test for hemolytic anemia, testing blood against commercial antibodies, and it is often used when diagnosing anemias, particularly in equine medicine used for diagnosis for Neonatal Isoerythrolysis. The Coombs Test will often be positive post transfusion, so it should be interpretedwith caution in horses who have already received blood products [59]. The JFAT is discussed in more detail in Section 7, however, it is a test for compatability between foal erythrocytes and maternal colostrum. The colostrum contains high concentrations of immunoglobulin, and if the mare is immunized against the foal’s erythrocyte antigen, colostral antibodies will induce agglutination when combined with the foals’ red cell portion [60].

### 4.6. Blood Volume Calculations

There are several different ways to calculate the volume of blood required by the recipient. The most accurate equation for required volume of transfusate considers both the donor and recipient PCV, the bodyweight in kg of the recipient, and a transfusion conversion factor, see Table 2. Other calculations give a more crude estimate but have reasonably good agreement in volumes required [61].

The volume of blood that can be taken from any one donor depends on the body weight of the donor. Circulating volume is approximately 8% of the animal’s bodyweight, and a maximum of 20% of the circulating volume can be collected if needed, however, more conservative collections of 15% are preferable, see Table 3. If the clinician decides to collect >10% of circulating volume from the donor then an equivalent volume of balanced electrolyte solution should be provided to restore circulating volume. Additionally, if a donor is being used repeatedly, less volume should be harvested to avoid detriment to the donor animal [8,44,49]. If excessive volumes of blood are harvested from the donor, then clinical signs of acute blood loss will become apparent.

### 4.7. Collection and Transfusion Practicalities

Clinicians who routinely perform blood transfusions will benefit from organizing a transfusion kit containing all the required items in one readily accessible location. There is some individual variation in preferred technique, however the authors recommend restraining the blood donor in stocks with a hay net hung above withers height. One jugular vein is clipped and aseptically prepared. The authors prefer to collect blood via a 12 g intravenous (IV) catheter, placed against the direction of blood flow (catheter tip towards the head), to facilitate rapid blood collection, and recommend blocking the skin with 2% Lidocaine or similar, to reduce aversion to IV catheter placement especially in horses used repeatedly as blood donors. Blood is collected into commercially available double port blood collection bags, pre-filled with 150 mL CPDA (citrate-phosphate-dextrose-adenosine) anticoagulant per 850 mL of blood. CPDA anticoagulant helps to prolong the shelf life and preserve the erythrocytes post-collection. The collected blood should be gently rocked during acquisition to prevent clotting and maintain even concentration of anticoagulant. Weighing the collected blood allows for the most accurate estimation of volume collected, as 1 L of blood weighs 1 kg.

Once blood is collected, the donor can be bolused with replacement fluids if deemed necessary. Collected blood can be administered immediately, or can be stored depending on the collection technique and sterility, for up to 35 days [62]. The blood can then be transfused to the recipient. Blood should be administered through a filtered blood administration set to remove clinically significant clots.

The recipient should be monitored closely for transfusion reactions, with a full physical examination including temperature, pulse, and respiration count (TPR) prior to initiation of transfusion. Blood should be administered at 1 drop per second for the first 5 min and if there are no changes in TPR values, the infusion rate can be increased as needed. Vital parameters should be monitored every 15 min throughout the transfusion.

#### Washed Red Cells

Washing red cells is a process where the clinician or laboratory removes the plasma component and adhered antigens from the red cell portion of the blood to eliminate the donor plasma antigenicity of the transfusion. It is only appropriate when the main requirement of transfusion is to restore PCV/oxygen carrying capacity, and is most commonly used in cases of NI, when the dam is the only available donor.

Blood is collected in CPDA or sodium citrate anticoagulated bags and either spun at low velocity or allowed to separate via gravity. The plasma portion is then removed, the erythrocytes resuspended in saline, and either administered or ideally washed a second time [63,64].

## 5. Contraindications and Complications of Transfusion

Blood transfusions are an accessible and lifesaving procedure for a variety of circumstances; however, they are not a procedure without risk. There are both immediate and long-term consequences to be considered when deciding if transfusion is indicated, and the clinician can mitigate the risks posed with a thorough knowledge of possible adverse events.

### 5.1. Transfusion Reactions

Transfusion reactions usually occur within the first 15 min of initiating administration of the blood, however some delayed reactions can occur in the hours following transfusion. Reactions are significantly more likely after multiple transfusions than after one single administration of exogenous blood [27,65]. Transfusion reactions can be divided into different categories, based on the onset of reaction, or pathological origin of the reaction. It is easy to remember the spectrum of transfusion reactions if approached from their mechanistic origin, as well as their clinical signs. Infusion of biological origin material can be associated with Type 1, Type 2 or Type 4 hypersensitivity reactions. Immediate hypersensitivity reactions are usually Type 1 hypersensitivity reactions, mediated by IgE and resultant histamine release, with clinical signs ranging from fulminant anaphylaxis to pyrexia, shivering, tachycardia, tachypnoea and urticaria. Frequently, elevated rectal temperature is the first sign of impending transfusion reaction, thus monitoring rectal temperature during blood administration is critical for safe transfusion [9,44]. Type 2 hypersensitivities are mediated by IgG or IgM and require prior exposure to incompatible blood. Type 4 hypersensitivity reactions occur most commonly after uncrossmatched or multiple transfusions. These are cell mediated reactions that occur up to 12–24 h post transfusion, and the most common clinical signs are urticaria and pruritis [66]. It is also pertinent to consider that the respiratory and gastrointestinal tracts are the shock organs of the horse and so diarrhea is one of the key early clinical signs of anaphylaxis in this species [67]. As such, acute onset diarrhea or colic during transfusion is an indication to stop administering the blood product and to treat the reaction with corticosteroids and/or epinephrine see Table 4 [68].

### 5.2. Incompatible Blood

Transfused erythrocytes have an approximately one week survival time in circulation, however Type II hypersensitivity reactions can occur, and lead immunoglobulin mediated to early and severe hemolysis [56,65]. If a horse has no previous exposure to biological products or blood, either administered by a veterinarian, due to breeding, gestation or parturition, and no accidental exposure via dirty medical equipment, the risks of a first unmatched blood transfusion are usually outweighed by the medical need for oxygen-carrying capacity. Subsequent transfusions of un-crossmatched blood, however, carry increasing risks of severe immunological reaction.

Other adverse outcomes of transfusion include sepsis from breaks in sterility, and bacterial contamination of the sample. Additionally, administration of blood products containing antibodies can, in rare cases, result in acute lung injury, referred to as TRALI—Transfusion Associated Acute Lung Injury, which occurs when donor antibodies are deposited in the microvasculature of the lungs, and leads to acute severe pulmonary edema [27,69,70]. In 1–3% of humans receiving whole blood transfusions, a syndrome of nonhemolytic febrile reactions occurs within the first 12 h post transfusion, due to the infusion of cytokine-producing leukocytes, which react with recipient blood and release inflammatory molecules. In human medicine, leukoreduction is a common process performed to reduce white blood cell concentration in stored whole blood, to minimize such reactions [65,71]. Leukoreduction is a common practice in equine autologous conditioned serum products to reduce reactions on infusion [72], however few operations have equine blood banks with facilities to leukoreduce whole blood, so it warrants noting that white cells infused with the required erythrocytes may bring complications of their own.

### 5.3. Long Term Complications

Iron overload, or hemochromatosis, is a sequela of multiple transfusions, particularly in foals where the milligrams of iron transfused per kilogram of bodyweight can become relatively much higher than in adult horses. As the transfused erythrocytes are broken down at a faster rate than autologous red cells, the iron handling capacity of the liver can become overwhelmed, leading to cytotoxic levels of unmetabolized iron in the liver, resulting in hepatocellular damage [51,73]. A sudden rise in Aspartate Aminotransferase (AST) following high volume blood transfusion, especially after 4 or more liters of blood have been given to a foal under 100 kg, should alert the clinician to hepatic damage and is associated with poor prognosis for survival [64,74].

## 6. Neonatal Alloimmune Disorders

Alloimmunity refers to an immune reaction against non-self-antigens from members of the same species. One of the most common alloimmune conditions in veterinary medicine is Neonatal Isoerythrolysis (NI) of equine neonates. This is a condition that occurs most often when a multiparous mare has developed an antibody population against a non-self-blood type. Alloimmune conditions are occasionally seen in primiparous mares who have a history of biological product administration. If the fetus/foal inherits a paternal blood type to which the mare produces antibodies, the dam’s colostrum will contain anti-erythrocyte antibodies which will induce an hemolytic anemia in the foal within days after birth [60]. About 1% of all foals born will experience NI if colostrum is not monitored correctly [60,75]. Foals born to multiparous mares, mares with a history of having an NI foal, or mares who are known to have received blood products, should receive a jaundiced foal agglutination test (JFAT) prior to being allowed to nurse the mare. The JFAT is a crude agglutination test performed with the mare’s colostrum and the foal’s erythrocytes. If agglutination is detected, the risk of NI is very high, and the foal should be prevented from nursing the mare until at least 24 h of age when the GI tract has ceased to translocate immunoglobulin efficiently. In these cases, plasma and/or donor colostrum must be sourced and administered immediately [60,76,77]. As such, this is one of the most common reasons in equine medicine when a clinician will be required to transfuse whole blood or washed red blood cells. See Section 4.6 for details of the procedure for washing red cells.

Definitive diagnosis is made by confirming an incompatible cross match between maternal serum and foal erythrocytes, or by performing a Coombs Test, which assesses agglutination of foal erythrocytes and maternal colostrum prior to nursing [75]. Treatment for NI hinges on preventing continued intake of colostrum if the foal is under 72 h of age, transfusing cross matched donor whole blood or washed maternal erythrocytes, providing supportive care and preventing complications such as kernicterus (bilirubin encephalitis), iron overload hepatitis, and sepsis [64,78]. Washing erythrocytes consists of collecting anticoagulated whole blood from the mare, centrifugation or gravity sedimentation of the collection, removal of the plasma, and resuspension of the erythrocytes in saline to remove the plasma component and associated plasma proteins [77,79]. This procedure can be repeated twice. Foals requiring greater than 4 L of whole blood have been shown to be 19.5 times more likely to develop liver failure and require euthanasia than foals requiring less than 4 L of blood [64]. In addition to NI, there is a much less common alloimmune condition in foals, where alloantibodies are ingested in the colostrum which attack the mucocutaneous junctions, platelets and neutrophils, resulting in a severe ulcerative dermatitis, thrombocytopenia and neutropenia. These foals can present with or without associated hemolytic anemia [80].

## 7. Donkey Factor

Donkeys possess a unique red blood cell antigen, known as ‘Donkey Factor’, not found in horses. For this reason, it is not advised to transfuse donkey blood into a horse, as this will induce hemolytic anemia. However, donkeys can safely tolerate transfusions of horse blood. While the need to transfuse between horses and donkeys is relatively rare, the main implication of Donkey Factor is the increased risk of NI in mares carrying mule foals. The incidence of NI in mules is approximately 10% [79], significantly higher than the reported 1% in horses. Mares that have carried multiple mule offspring are likely to have become immunologically reactive to Donkey Factor, meaning that their colostrum will contain hemolyzing antibodies against the inherited donkey erythrocytes [77,81]. For these reasons, all mule neonates born to multiparous mares should be cross matched or have a colostral/erythrocyte coagulation test performed before being allowed to nurse.

## 8. Conclusions

In summary, blood transfusion is an approachable and potentially lifesaving technique, which can be performed safely and with significant benefit by equine practitioners in a variety of settings. Knowing and understanding the risks and benefits allows clinicians to make evidence-based decisions for the benefit of their patients, while managing and treating complications that may arise. Specific precautions should be taken when considering transfusions in neonates and non-horse equids.

## Figures and Tables

**Figure 1 animals-12-02162-f001:**
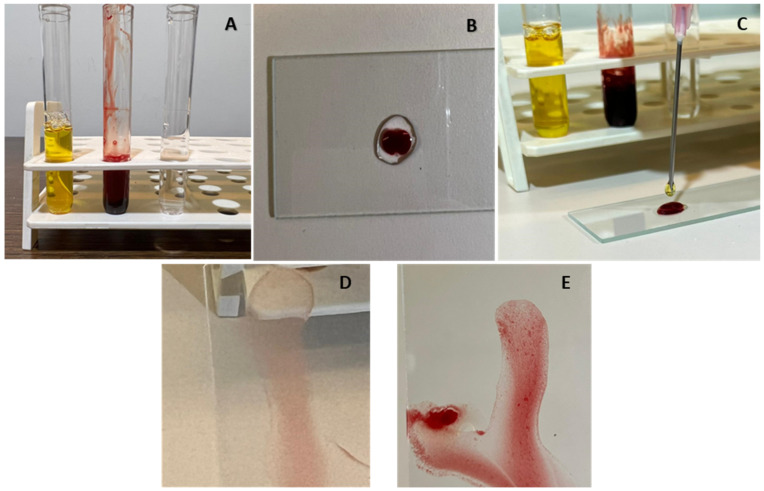
Crude slide agglutination test for blood type compatibility. Major cross match demonstrated. (**A**) Recipient serum separated from erythrocytes, donor erythrocytes separated from serum, and saline. (**B**) Amount of 1 drop saline and 1 drop donor erythrocytes, placed onto a glass slide. (**C**) Addition of 1 drop of recipient serum, the slide gently rocked until the three components were well combined. (**D**) No agglutination observed—compatible. (**E**) Agglutination grossly apparent in the mixed sample—incompatible.

**Table 1 animals-12-02162-t001:** Indications for blood transfusion in horses [8].

• Tachycardia
○ The higher the heart rate (HR) the more urgent
○ Severe bleeding may be accompanied by HR > 100 beats/min
• Tachypnoea
○ As above
• Decreased pulse quality
○ Thready pulses
○ Hard or impossible to palpate
• Cool extremities
• Pale mucus membranes
• Mentation changes
○ Anxiety
○ Distress
○ Depression
○ Compulsive thirst
• Hyperlactatemia
○ Serial sampling most helpful
○ Progressive increase indicates decreasing perfusion
• Decreased PCV
○ Acute drop of 10%
○ Absolute PCV < 12–15% usually requires
○ ±Decreased TS

**Table 2 animals-12-02162-t002:** Reference formulae for calculating required blood volumes.

$Transfusion\;Volume\;\left(mL\right)$
$=\frac{\left[Desired\;PCV-Actual\;PCV\right]}{Donor\;PCV}\times 90\times BW\left(kg\right)$
Alternate Calculations:
$1\%\;PCV\;Increase=1.5-2.2\times BW\left(kg\right)$

**Table 3 animals-12-02162-t003:** Volumes and frequencies of blood that can be safely harvested from donor horses without detriment to donor.

Blood Volume (BV)—8% BW (kg)
20% of BV
Example:
450 kg horse-circulating volume = 36 L
20% of 36 L = 7.2 L
Alternately:
10% BV every 4 weeks
7.5% BV every 7 days
1% BV every 24 h

**Table 4 animals-12-02162-t004:** Emergency Drug Doses.

Drug	Dose	Dose per 450 kg Horse
Dexamethasone 2 mg/mL	0.02–0.1 mg/kg IV	4–20 mL IV once a day
Prednisolone 50 mg/mL	2–5 mg/kg IV	18–45 mL IV once a day
Epinephrine/Norepinepherine 1:1000 (1 mg/mL)	0.01–0.02 mg/kg IV-(anaphylaxis) up to 0.5 mg/kg for asystole	4–9 mL IV repeated up to 3 times—can increase dose up to 225 mL for asystole

## Data Availability

Not applicable.

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
