# Peer review of "Blood Transfusion in Equids—A Practical Approach and Review"

_animals, 2022, doi:10.3390/ani12172162_

Round 1

Reviewer 1 Report

The submission refers to a review report form “Transfusion medicine in equids”, an important topic for equine medicine, however, for this manuscript publication I recommend the following corrections:

1 - The topics 2 and 3 must be improved and rewritten. For example: Topic 2 – Pathologies requiring transfusions: the authors listed whole blood loss, anemia, spleen. These topics are not related to pathologies requiring transfusions. To be more precise, the spleen is not a pathology! It is an organ. The first topic should be better explained, and the next ones are all related each other. The next topic “Surgical bleeding” should be included in the first topic.

2 – Line 56: infection agents must be used instead of viral infection, as a cause of anemia (some protozoan may also cause anemia). Recommendation: review causes of anemia in horses.

3 – Lines 139 – 144: I totally disagree “if the clinician has time to acquire minimum database…” It should be mandatory to have all those information related to hematological profile and infections disease’s status from donor before use its blood for transfusion.

Author Response

Dear Reviewer, 

Thank you very much for the feedback on this manuscript. It has sparked great improvement and highlighted key areas to be addressed. Please see below for specific responses to individual points.

Your time and attention is much appreciated. 

Sincerely, 

The Authors

1 - The topics 2 and 3 must be improved and rewritten. For example: Topic 2 – Pathologies requiring transfusions: the authors listed whole blood loss, anemia, spleen. These topics are not related to pathologies requiring transfusions. To be more precise, the spleen is not a pathology! It is an organ. The first topic should be better explained, and the next ones are all related each other. The next topic “Surgical bleeding” should be included in the first topic.

Thank you for this, the manuscript has been re-organized as suggested. 

2 – Line 56: infection agents must be used instead of viral infection, as a cause of anemia (some protozoan may also cause anemia). Recommendation: review causes of anemia in horses.

Excellent point thank you, this paragraph has been expanded upon and restructured to provide more detail and comprehensive information. 

3 – Lines 139 – 144: I totally disagree “if the clinician has time to acquire minimum database…” It should be mandatory to have all those information related to hematological profile and infections disease’s status from donor before use its blood for transfusion.

This point has been adapted and modified. I personally feel that there are situations in which the lifesaving nature of providing oxygen carrying capacity, especially in field situations, may outweigh the risks of un-screened donor blood, however these are few and far between. The risks of this have been stressed and the implications for such have been clarified as severe. 

Reviewer 2 Report

Please see separate word document.

Round 2

Reviewer 2 Report

The manuscript is significantly improved. There are 2 typos in the abstract. My main concern remains with the organization of the first section - disorders require transfusion. I would recommend utilizing a text book that describes causes for anemia or blood loss. Please reorganize this chapter so it makes sense (inlcuding adding surgical blood loss in with whole blood loss. why do hemostatic disorders have their own subcategory? why not trauma? why is GP mycosis mentioned and not uterine artery rupture? i think it should be trauma, surgeyr, coagulopathies in this first section; Same for anemia, why does anemia of chronic disease have a subcategory and other more important causes for anemia not... . If you discuss EPO make sure to also discuss rhEPO so as not to confuse anyone. If Table 1 is from Ref 8, please make sure to mention in in the Table 1 legend. I would suggest not including an actual HR because it is dependent on so many other factors...  Focus on a constellation of clinical (and laboaryr findings). It would be nice in this chapter to include different transfusion triggers for acute hemorrhage vs. chronic.

Finally, 
